Study on the influence of TRX suspension training on the specific balance ability of surfers

Wang Zhaoyi 1 2
Ma Yong mayong@whsu.edu.cn 1 2 3 4
Huang Qian 1 2
Guo Zhihao 1 2
Ja Mengyao jmy15737132785@163.com 1 2
Zheng Weitao 1 2
1 School of Intelligent Sports Engineering, Wuhan Sports University , Wuhan , Hubei , China
2 Key Laboratory of Sports Engineering of General Administration of Sport of China, Wuhan Sports University , Wuhan , Hubei , China
3 Specialised Research Centre for High-Quality Development of Competitive Sports, Wuhan Sports University , Wuhan , Hubei , China
4 Engineering Research Center of Sports Health Intelligent Equipment of Hubei Province, Wuhan Sports University , Wuhan , Hubei , China
Climstein Mike
Electronic publication date: 2025 Oct 16
Publication date: 2025
Volume: 13
Electronic Location ID: e20180
Received 2024 Oct 31; Accepted 2025 Sep 12
Copyright: ©2025 Wang et al.
Copyright year: 2025
Copyright holder: Wang et al.
License: This is an open access article distributed under the terms of the Creative Commons Attribution License, which permits unrestricted use, distribution, reproduction and adaptation in any medium and for any purpose provided that it is properly attributed. For attribution, the original author(s), title, publication source (PeerJ) and either DOI or URL of the article must be cited.
License URL: https://creativecommons.org/licenses/by/4.0/

Keywords: Suspension training, Traditional balance training, Balance board lateral squat, Surf-specific balance ability, Growth rate, Intervention training

Funding: The East Lake Scholars Sponsorship Program of Wuhan Sports University in China, Science and Technology Team Foundation of Wuhan Sports University 21KT02 The 14th Five-Year-Plan Advantageous and Characteristic Disciplines (Groups) of Colleges and Universities in Hubei Province 2021-05 This work was funded by the East Lake Scholars Sponsorship Program of Wuhan Sports University in China, Science and Technology Team Foundation of Wuhan Sports University (grant number 21KT02), the 14th Five-Year-Plan Advantageous and Characteristic Disciplines (Groups) of Colleges and Universities in Hubei Province (grant number 2021-05). The funders had no role in study design, data collection and analysis, decision to publish, or preparation of the manuscript.

==============================
Objective

To investigate the effects of total resistance exercise (TRX), suspension training on the specialized balance ability of surfing athletes.

Methods

A total of 32 Chinese National Surfing Team athletes were randomly assigned to TRX suspension training group (TRX group) and traditional balance training group (TB group), both undergoing an 8-week intervention training program consisting of three sessions per week lasting approximately 30 minutes each to improve balance ability. The balance board lateral squat specialized balance ability test was administered at different intervention phases to examine changes in participants’ specialized balance ability.

Results

Following the 8-week intervention, both training modalities demonstrated significant improvements in surfing specialized balance metrics relative to baseline (p < 0.01 for all comparisons). Inter-group comparisons revealed statistically superior outcomes for TRX suspension training, with marked divergence emerging at the 5-week assessment (p = 0.005 < 0.01, |d| = 1.062 > 0.8) that intensified through intervention completion (p = 0.000 < 0.01, |d| = 1.417 > 0.8).

Conclusion

Both 8 weeks of TRX suspension training and traditional balance training were effective in enhancing the specialized balance of surfers, but TRX suspension training was superior to traditional training in terms of enhancement, and this advantage was significantly demonstrated after 5 weeks and continued until the end of the training. TRX suspension training aligns with the balance demands of surfing, making it an effective training method. It is feasible for coaches to formulate customized TRX suspension training regimens tailored to surfers’ proficiency levels and core stability, integrating sport-specific movement patterns to enhance athletes’ neuromuscular adaptations for improved biomechanical control during dynamic wave navigation.

Introduction

Surfing, a widely popular and highly competitive sport, requires surfers to possess refined physical fitness, technical proficiency, and tactical awareness for success (Wang et al., 2024; Mendez, Bishop & Hamer, 2006; Secomb, Sheppard & Dascombe, 2015a; Anthony & Brown, 2016a). Specifically, aerobic capacity, anaerobic capacity, intermittent endurance, and upper body strength are critical for paddling through waves (Lundgren et al., 2014; Donaldson et al., 2022), while balance, flexibility, agility, and lower body coordination enable efficient wave navigation (Frank et al., 2009; Secomb, Sheppard & Dascomb, 2015b). When combined with optimal technical and tactical knowledge, these physical attributes allow athletes to execute more challenging maneuvers and achieve higher scores (Sheppard et al., 2013; Forsyth et al., 2017). Current research on surfing primarily focuses on technique analysis (e.g., paddling skills) (Farley et al., 2016; Coyne et al., 2017), competition performance evaluation (Ferrier et al., 2018; Ma et al., 2023), technical and tactical assessment (Forsyth et al., 2021), and surf-specific physical training (Ma & Xu, 2020; Parsonage et al., 2020).

Each sport demands sport-specific training approaches, and surfing is no exception. Designing effective training protocols for surfers is challenging due to the dynamic and unpredictable nature of ocean conditions, necessitating flexible and adaptive strategies to develop surfing-specific skills and fitness (Monaco et al., 2022). To address these challenges, researchers and coaches have explored diverse training methods. For example, Farley et al. (2016) demonstrated that a 5-week high-intensity interval training and sprint interval training program improved the aerobic and repeated sprint paddling capacities of adolescent surfers. Coyne et al. (2017) found that a 5-week upper body maximal strength training program only slightly enhanced paddling efficiency. Additionally, Axel et al. (2018) showed that an 8-week, twice-weekly core strength training program improved core muscle strength, power, countermovement jump performance, maximal power output, and rotational flexibility in adolescent competitive surfers.

Contemporary kinesiological studies indicate that the dynamic oceanic conditions in surfing—characterized by transient wave formations and fluid dynamics—make postural stability a critical biomechanical determinant of surfing proficiency (Alcantara, Prado & Duarte, 2012; Anthony et al., 2016b; Dowse et al., 2021). Superior balance control is a fundamental prerequisite for success in surfing (Guo & Wang, 2022). Given the marine environment, unstable surface training has become an essential approach, with equipment such as BOSU balls, balance boards, stability pads, and stability balls used to strengthen core muscles, improve balance and proprioception, and enhance athletic performance.

In surfing research, Tran et al. (2015) compared the effects of resistance training on strength and sensorimotor abilities (the coordination of sensory input and motor output) in adolescent surfers under stable versus unstable conditions. The findings showed that, compared to traditional stable resistance training, unstable training did not provide a significant advantage in enhancing strength, power, or sensorimotor skills. Despite evidence that stable and unstable training have similar impacts on surfers’ balance and strength, further exploration of targeted training methods is needed to optimize performance. Notably, few studies have investigated balance training aligned with surfing’s athletic characteristics, and evidence-based recommendations for performance enhancement through surfing-specific movements remain limited (Forsyth et al., 2020; Monaco et al., 2023).

Total resistance exercise (TRX), a form of suspension training, utilizes body weight as resistance. Through closed-chain movements with suspension equipment, it maintains the body in a continuous unstable state, which helps improve balance ability (Dong, Xue & Zeng, 2019; Wei et al., 2009). During training, athletes must fully engage proprioceptive abilities to activate neuromuscular and motor responses (Campa, Silva & Toselli, 2018; Demirarar et al., 2021). This approach facilitates central nervous system adaptation to instability, optimizes neuromuscular recruitment and regulation, and thereby enhances athletic performance and balance—benefits not achievable with traditional training methods (Li et al., 2008).

TRX suspension training is widely applied across sports. For example, hurdle runners and swimmers showed improved performance on the eight-level plank test and single-leg stance test with eyes closed after suspension training, indicating enhanced core stability and balance (Meng, 2020; Wang, Zhou & Lu, 2012). Similarly, tennis and volleyball players exhibited reduced deviation distances and sway amplitudes during single-leg support post-training, supporting TRX’s role in improving body control and balance (Fu & Li, 2014; Sun, Gao & Sun, 2010). Additionally, TRX training has enhanced gymnasts’ stability and technical proficiency during rotational movement landings (Li et al., 2010; Cao, 2021) and curling athletes’ accuracy and balance during performance (Liu, Xu & Zhang, 2018). Compared to traditional training, 8 weeks of suspension training yielded greater improvements in Taekwondo athletes’ eight-level plank test and 20-second controlled leg kick test performance, highlighting its efficacy in enhancing core stability and sport-specific skills (Tian & Fu, 2024). Twelve weeks of TRX training further improved core stability and strength in aerobics athletes (Pang, 2023). Traditional balance training, typically conducted on stable surfaces, has limitations in activating and strengthening core and supporting muscles, making it less effective for comprehensively enhancing balance, coordination, and strength (Qiao & Yuan, 2010). Although TRX suspension training demonstrates faster and more significant effects than traditional training—especially in improving core stability, balance, and technical precision—its efficacy for surf-specific balance remains underexplored.

TRX suspension training offers a non-stationary movement environment similar to surfing and effectively enhances neuromuscular control, coordination, and surf-specific balance, making it highly suitable for surfing balance training. Currently, coaches of the Chinese National Surfing Team have observed that traditional balance training has reached a plateau with limited room for further performance improvement. This research gap necessitates systematic exploration of surf-specific balance training protocols to meet the complex balance demands of wave riding. In response, national team coaches initiated this study to identify more appropriate training methods, aiming to effectively improve athletes’ balance ability, enrich surf-specific balance training practices, and contribute to enhanced surfing performance.

Given the importance of dynamic balance in surfing, we hypothesized that surfers with high score in the balance board lateral squat test would exhibit greater technical proficiency and fewer errors during actual surfing.

Materials & Methods

Participants

Sample size was determined using G*Power 3.1.9.2 software (Heinrich-Heine-Universität Düsseldorf, Düsseldorf, Germany). Based on prior balance training studies in athletic populations (Wang et al., 2023), a medium effect size (d = 0.5) was adopted. With a significance level (α) set at 0.05 and statistical power (1−β) at 0.80, the calculation indicated a minimum requirement of 26 participants to detect meaningful effects. To account for potential attrition, all athletes from the Chinese National Surfing Team were initially screened for eligibility. Inclusion criteria were: (a) normal motor function confirmed by pre-experimental physical examination, with no current sports injuries or acute illnesses; (b) competitive performance at the national level, defined as individual rankings in the top 12 or team rankings in the top 8 at major national events (including the National Games and National Championships). The exclusion criteria were: (a) history of orthopedic surgeries involving the knee or ankle, fractures, or other significant lower extremity musculoskeletal injuries; (b) lower extremity injuries within the preceding 3 months, or any existing pathology potentially impairing balance function. After screening, 36 eligible athletes from the Chinese National Surfing Team were enrolled. During the 8-week intervention, four participants were excluded due to incomplete training attendance, leaving 32 athletes with 100% training adherence for final analysis. This cohort comprised 19 national elite athletes (representing China’s highest competitive standard, comparable to international levels) and 13 first-class athletes (recognized as high-caliber competitors within the national sports framework).

A single-blind design was employed in this study, with participants blinded to their group assignment throughout the intervention to minimize bias from subjective expectations or psychological factors. To ensure balance in key baseline characteristics—including sex, age, height, weight, and competitive level—block randomization was implemented. Participants were first stratified into blocks based on these variables; within each block, assignments to the two experimental groups were generated using a computerized random number sequence. Participants were numbered within blocks and allocated sequentially according to the random sequence, ensuring comparability of baseline characteristics across groups.

Group allocation, outcome measurements, and blinding procedures were conducted independently by a researcher who remained unaware of participants’ group assignments throughout the study, thereby reducing measurement bias and subjective influence. All allocation records were securely stored and only unblinded by designated personnel during data analysis, ensuring strict allocation concealment to enhance study credibility and transparency.

To further mitigate potential bias—particularly for outcomes with subjective components—outcome assessors were also blinded to group assignments. Assessors were not informed whether participants belonged to the TRX suspension training group or the traditional balance training (TB) group during data collection, and this blinded status was maintained until all outcome data had been fully collected and initially recorded.

These rigorous methodological measures—including randomization, participant blinding, allocation concealment, outcome assessor blinding, and balanced control of key baseline characteristics—collectively enhanced the scientific validity of the experimental design and the reliability of the results. Participants were ultimately randomized to either the TRX suspension training group (TRX group, n = 16) or the traditional balance training group (TB group, n = 16). The CONSORT flow diagram is presented in Fig. 1, and baseline characteristics of the grouped participants are summarized in Table 1.

Figure 1 Flowchart of participants.

All participants provided written informed consent, and the study was approved by an informed consent form signed by all participants and by the Ethics Committee of Wuhan Sports University (No. 2021030). The study confirmed that all methods were performed in accordance with relevant guidelines and regulations.

Research process

Experimental program

This study was conducted during the 2022 National Surfing Team training period. As participants were professional athletes, no medications or comorbidities that could potentially influence study outcomes were reported. All aspects of training, diet, and work-rest schedules were standardized and centrally managed to ensure consistency. The experiment was conducted in the Strength and Conditioning Room of the Chinese National Surfing Team, located in Lingshui Autonomous County, Hainan Province, China. All training activities—including fitness sessions and surfing practice—were uniformly scheduled and overseen by the national team coaching staff to minimize confounding variables. To control for the potential influence of concurrent surfing practice, all participants followed identical surfing training protocols throughout the 8-week intervention period. This centralized management ensured uniformity across all aspects of participants’ routines, with the sole variable being the type of balance training (TRX vs. TB) administered to each group.

The study adopted a “1+8” experimental schedule, consisting of a 1-week adaptation phase followed by an 8-week formal intervention. The adaptation phase aimed to ensure athletes could perform training movements with accuracy and standardization, establishing a foundation for the formal experiment. The formal intervention consisted of 8 weeks of training, conducted 3 times per week with each session lasting approximately 30 min (Wei, Wang & Zhao, 2022). Both groups focused on balance and core stability training throughout the intervention. During the formal phase, a professional coaching team (comprising two coaches with extensive sports training experience) provided full-time supervision. Coaches conducted real-time on-site observation of athletes’ movement execution in each session, offering individual guidance and corrections to ensure movement standardization and accuracy. Additionally, athletes’ acceptance of training movements and preliminary responses to training were continuously monitored throughout the phase.

Table 1 Baseline characteristics of study participants (sex, number, age, weight, height and years of training).

	Sex & Numbers	Age/y	Height/cm	Weight/kg	Training years/y	
TRX	8M & 8F	15.38 ± 2.15	164.69 ± 7.78	52.81 ± 7.64	2.88 ± 0.48	
TB	8M & 8F	15.44 ± 2.09	164.69 ± 7.24	52.31 ± 8.08	2.88 ± 0.70	
p	–	0.936	1.000	0.863	1.000	
Cohen’s d		−0.028	0.000	0.064	0.000	
Notes.

Height and weight collection equipment for height and weight tester (Kefun Group, Model: KF-1328, Zhejiang, China). Measuring range: 0.1–180 KG, 70–190 CM.

TRX TRX suspension training group

TB Traditional balance training group

M Males

F Females

Training load was individualized based on each athlete’s optimal intensity (repetition count or duration) for standardized movements, with adjustments made according to real-time physical status. This ensured comparable overall load between groups while adhering to the principle of gradual progression (Wang & Li, 2007). Training intensity was prescribed within a target range (rather than mandatory fixed quantities or durations), and coaches used a standardized checklist to verify exercise fidelity, with minor adjustments permitted within the defined intensity parameters.

Training adherence was monitored through multiple methods: athletes completed post-session training logs documenting completed content, subjective feedback, and physical conditions; coaches recorded attendance before each session and evaluated in-session performance. Athletes exhibiting insufficient effort or non-compliance received timely communication from coaches to address underlying issues and provide targeted supervision.

Variations in training intensity and program progression across the 8-week intervention are detailed in Table 2.

Table 2 Training intensity progression chart (8 weeks).

Training phase	Week	Number of groups	Time/Reps	Training frequency	Notes	
E1	1	4	20–40s (each side) 60s rest between sets	3 times/week 30 min/times	• Static training
• Adapt to the training pace
• Master the correct form	
2	
E2	3	30–50s/12–16 reps
(each side) 60s rest between sets	• Combine dynamic and static training
• Increase training volume and intensity	
4	
5	
E3	6
7
8	40–60s/16–50 reps
(each side) 60s rest between sets	• Dynamic training exercises
• Maximize training volume and intensity
• Strengthen and improve balance ability	

Training contents

Training protocols for both groups were developed through rigorous deliberations and evidence-based research by an expert panel, which included elite coaches and strength and conditioning specialists. This collaborative approach ensured the scientific validity, efficacy, and close alignment of the training methods with the technical demands specific to surfing. For the TRX group, the expert panel formulated training movements and protocols by integrating surfing’s biomechanical characteristics, drawing on best practices from analogous sports (e.g., skateboarding, freestyle skiing), and synthesizing successful applications of suspension training across diverse athletic disciplines. In contrast, the traditional balance (TB) group followed the established balance training protocol routinely implemented by the Chinese National Surfing Team. This long-standing protocol provided a valid comparative baseline to assess the relative efficacy of TRX suspension training in enhancing balance ability. This controlled experimental design enabled the study to validate both the innovation and applicability of TRX training for surfing, while also offering evidence-based insights to optimize practical physical training regimens in the sport.

The 8-week training was divided into three phases: (E1) Basic adaptation (weeks 1–2), (E2) Quality enhancement (weeks 3–5), and (E3) Consolidation and reinforcement (weeks 6–8) (Fu & Li, 2014). Training intensity and difficulty were designed to follow a gradual, incremental progression throughout the intervention. During the entire training period, athletes were restricted to performing only the movements specified in the program, with no additional balance training exercises introduced. In the E1 phase, both groups focused on static training movements to help athletes acclimatize to the initial training rhythm (Meng, 2020). At this stage, training difficulty was set at a low level, with emphasis placed on guiding athletes to familiarize themselves with the basic sequence and technical specifications of the training movements. For static exercises, athletes were typically required to maintain a specific posture or position for a predetermined duration. The TRX group’ s training movements were: suspended prone planks, suspended legs supine hip bridge, suspended lateral one-handed brace and suspended plank balance (Fig. 2). The main training contents of the TB group’ s main training contents were: supine leg raises, plank support, hip bridge and lateral bridge (Fig. 3). The number of sets of movement training in this phase were all 4 sets, the duration of each set was 20–40 s, and the inter-set interval was 60 s (Wang & Li, 2007).

Figure 2 TRX group (E1) basic adaptation phase training movements diagram.

Figure 3 TB group (E1) basic adaptation phase training movements diagram.

In the E2 phase, both groups integrated static and dynamic training modalities, with a marked increase in training load and difficulty compared to the previous phase. For the static training component, the duration of posture maintenance was extended beyond the E1 phase requirements. The introduction of dynamic training exercises further enhanced the complexity of the training regimen. The training movements in the TRX group included: suspended prone legs supported open and close, forward and backward movement of the suspension plate support, suspended single leg V-support, suspended prone support group and suspended straight leg flexion (Fig. 4). The training movements in the TB group included: plate support movement, knee rolls, straight legged foot rolls, lateral rolls and air cycling (Fig. 5). The number of training sets of movements in this phase were all 4 sets, and the inter-set interval was 60 s. The duration of each set of static movements was 30–50 s, and the dynamic type of movements was 12–16 repetitions (Wang & Li, 2007).

Figure 4 TRX group (E2) quality enhancement phase training movements diagram.

Figure 5 TB group (E2) quality enhancement phase training movements diagram.

In phase E3, both groups focused exclusively on dynamic training maneuvers, with training difficulty and load undergoing another substantial increase compared to prior phases. Concurrently, the complexity of dynamic training movements saw a significant advancement, incorporating a greater number of simulations of real-world surfing techniques. The TRX group consisted of the following training movements: suspended single-legged double-armed support prone leg tucks, suspended single-legged one-handed lateral support leg suction, suspended single-legged prone support lateral knee lifts, suspended prone support knee lift runs and Suspended prone one-handed rotations (Fig. 6). The TB group consisted of the following training movements: supine chin ups, single arm support turn, one-handed prone knee lifts, group body movements and prone mountain running (Fig. 7). The number of training sets for all movements in this phase was 4, the interval between sets was 60s, and the movements were 16–20 repetitions (or 40–60 s) per set (Wang & Li, 2007).

Figure 6 TRX group (E3) consolidation reinforcement phase training movements diagram.

Figure 7 TB group (E3) consolidation reinforcement phase training movements diagram.

During training, coaches dynamically adjusted training intensity based on real-time feedback from participants, ensuring that training loads remained within the predefined thresholds of the program. When participants reported excessive intensity, coaches implemented adjustments by reducing movement difficulty, lowering training volume, or extending rest intervals. Conversely, if participants perceived intensity as insufficient, coaches increased movement complexity, elevated training resistance, or shortened rest periods. This individualized adjustment strategy ensured that training outcomes aligned with program objectives while accommodating participants’ actual physical conditions.

Test indicators

All tests were conducted by the same researcher in a fixed location: the Strength and Conditioning Room of the Chinese National Surfing Team in Lingshui County, Hainan Province, China. Testing was standardized under controlled environmental conditions (temperature: 20–27 °C; humidity: 68%–78%; quiet indoor setting) and scheduled within a consistent time window (7:00–10:00 PM). To eliminate confounding effects of acute fatigue, all tests were administered at least 48h after the last training session. To prevent inter-participant interference, assessments were conducted individually using a sequential entry protocol, where each participant entered the testing area only after the previous participant had completed all tests. The testing schedule was divided into four sessions aligned with key training milestones, as detailed in Table 3.

The primary objective of this study was to investigate the effects of TRX suspension training on surfers’ specialized balance abilities, making the selection of valid assessment tools for this construct critical. Unlike sports evaluated through objective metrics such as time or distance, surfing performance assessment has traditionally relied heavily on subjective rating scales (Monaco et al., 2023). However, subjective outcome measures are inherently limited by potential biases in perceptual interpretation (Pollard, Johnston & Dixon, 2007).

To address this limitation, researchers have increasingly utilized land-based simulations to replicate surfing-specific demands (Mendez, Bishop & Hamer, 2006; Liu, 2015). Building on this approach, the current study advanced existing simulation protocols—specifically extending the use of Indo Boards for training the pop-up phase of surfing (Pérez-Gutiérrez, Castanedo & Cobo, 2023)—and integrated insights from practical training and evaluation scenarios within the Chinese National Surfing Team. Ultimately, the balance board lateral squat test was selected as the primary outcome measure. This tool aligns with the sport-specific characteristics of surfing, is easy to administer, and involves straightforward operation. It has been widely implemented in internal training programs of the Chinese National Team and has demonstrated utility in evaluating athletes’ balance capabilities.

The balance board lateral squat test effectively evaluates athletes’ dynamic balance and lateral movement capabilities, which closely align with the practical demands of surfing. Compared to traditional static balance assessments (e.g., the single-leg stance test), this test more accurately reflects athletes’ balance control in dynamic scenarios. The test protocol involves a sequence of movements on the balance board: standing, squatting, touching the board’s side, and returning to a standing position (Fig. 8). This sequence closely mimics the squatting, bending, and rising motions involved in surfers’ “take-off” phase, as well as technical maneuvers such as bottom turns and cutbacks—critical actions in surfing that require precise balance control during intricate movements. These surfing-specific actions demand athletes to maintain dynamic balance in an unstable aquatic environment, placing stringent requirements on lower-limb strength, core stability, and proprioceptive function.

Table 3 Four testing time points schedule.

Organization of the training phase	Testing time	
E0	Before the experiment starts	
E1	Test after completion of the Basic Adaptation Phase (1–2 weeks)	
E2	Test after completion of the Quality Enhancement Phase (3–5 weeks)	
E3	Test after completion of the Consolidation and Strengthening Phase (6–8 weeks)	

Figure 8 Schematic diagram of the standard action in the balance board lateral squat test.

The balance board lateral squat test has been integrated into the practical training programs of the Chinese National Surfing Team and is recognized as a valid instrument for assessing athletes’ balance abilities. It is important to acknowledge that surfing’s unique characteristics necessitate a combination of water-based and land-based simulation approaches for comprehensive balance assessment. The highly dynamic and unstable nature of the aquatic environment makes it challenging for traditional ground-based balance tests to fully replicate real-world surfing conditions. As a land-based simulation tool, the balance board lateral squat test can, to a certain extent, reflect athletes’ balance capabilities relevant to surfing performance.

Evaluation of test effect: Within the designated time frame, a higher number of completed balance board lateral squats indicates superior ability to control body posture and maintain balance under unstable conditions. This performance metric also reflects an athlete’s capacity to adjust body positioning on waves more rapidly, precisely, and efficiently.

Test procedure: Upon the tester’s command “Begin,” a one-minute countdown timer is initiated. The athlete immediately steps onto the balance board, assumes a sideways standing position to establish balance, then performs a squat to touch the board. The following technical requirements must be met:

(a) Squat depth: The thigh of the front leg must reach at least a position parallel to the ground during the squat phase.

(b) Hand contact: Both hands must touch the board simultaneously; attempts with only one hand touching or asynchronous hand contact are not counted.

(c) Falls: If an athlete falls off the board during testing, they may remount and resume the test immediately after readjusting their position.

(d) Standing phase: When rising from the squat, the athlete must return to a full upright standing position to qualify as a completed repetition.

Figure 8 illustrates the standardized protocol and technical specifications for a single repetition in the Balance Board Lateral Squat test. The figure clearly depicts the correct posture and range of motion of body segments across all phases of the squat movement, serving as a precise reference for ensuring consistent and accurate test administration.

Statistical analyses

All data were analyzed using SPSS statistical 26 software (Version 26.0, IBM Corporation, Armonk, NY, USA) and presented as mean ± standard deviation (X ± SD). Prior to formal analyses, the Shapiro–Wilk test was applied to assess data normality, and subsequent statistical procedures were conducted only after confirming that the data followed a normal distribution. A homogeneity test was performed on baseline characteristics and balance ability metrics between the two groups to verify the validity of randomization. Within-group comparisons of test results before and after the intervention, as well as across training phases, were conducted using paired-sample t-tests. Between-group comparisons were analyzed using independent-sample t-tests. Effect sizes were quantified using Cohen’s d to evaluate the magnitude of differences.

Statistical significance was defined as follows: non-significant (p > 0.05), significant (p < 0.05), and highly significant (p < 0.01). Effect size thresholds were categorized as small (|d— = 0.2), medium (|d| = 0.5), and large (|d| = 0.8).

Results

Homogeneity test

To assess the comparability of baseline characteristics and surf-specific balance abilities between groups prior to the intervention, relevant indicators were measured in both athlete groups. Statistical analysis revealed no significant differences between the two groups in physical characteristics (Table 1) or surf-specific balance abilities (Table 4) (all p > 0.05, |d| < 0.2). These findings confirm the baseline comparability of the two groups.

Table 4 TB group balance board lateral squat test results.

	TRX	TB	t	p	Cohen’s d	
Balance board lateral squat (pcs)	29.06 ± 4.05	29.38 ± 4.18	−0.208	0.837	−0.075	

Variations in specialized balancing ability in the two groups

As shown in Tables 5 and 6, both groups demonstrated highly significant improvements in balance board lateral squat ability following 8 weeks of training (all p = 0.000 < 0.01, |d| > 0.8).

Table 5 Organization of the four tests.

Phase	Mean value difference (pcs)	Growth rate (%)	t	p	Cohen’s d	
E0 → E1	7.50	25.81	−11.619	0.000**	−1.550	
E0 → E2	13.81	47.53	−17.563	0.000**	−2.989	
E0 → E3	21.06	72.47	−24.670	0.000**	−4.402	
E1 → E2	6.31	17.27	−10.580	0.000**	−1.209	
E1 → E3	13.56	37.09	−18.155	0.000**	−2.528	
E2 → E3	7.25	16.91	−17.541	0.000**	−1.403	
Notes.

** where ** indicates a highly significant difference from the previous test result within the group, p < 0.01.

Table 6 TB group balance board lateral squat test results.

Phase	Mean value difference (pcs)	Growth rate (%)	t	p	Cohen’s d	
E0 → E1	4.31	14.68	−13.800	0.000**	−1.047	
E0 → E2	8.81	30.00	−21.469	0.000**	−2.189	
E0 → E3	14.31	48.72	−24.274	0.000**	−3.596	
E1 → E2	4.50	13.36	−14.863	0.000**	−1.181	
E1 → E3	10.00	29.69	−20.342	0.000**	−2.657	
E2 → E3	5.50	14.40	−10.488	0.000**	−1.503	
Notes.

** where ** indicates a highly significant difference from the previous test result within the group, p < 0.01.

In the TRX group, the mean score of the balance board lateral squat test increased by 21.06 units post-intervention, corresponding to a growth rate of 72.47%. In the TB group, the mean score of the same test increased by 14.31 units, with a growth rate of 48.72%.

Across all test phases, both groups exhibited significant improvements compared to pre-intervention measurements (all p = 0.000 < 0.01, |d| > 0.8).

Comparison of specialized balancing abilities

Independent samples t-test results indicated no significant difference in training effects between the two groups after the initial 2-week intervention (p = 0.095 > 0.05, |d| = 0.192 < 0.2), despite both groups achieving significant improvements in test scores.

Following 5 weeks of intervention, a highly significant difference emerged between the two groups (p = 0.005 < 0.01, |d| = 1.062 > 0.8). Prior to intervention, the mean number of squats was 42.88 in the TRX group and 38.19 in the TB group, with a mean difference of 4.69 squats between groups. After 8 weeks of intervention, the mean number of squats increased to 50.13 in the TRX group and 43.69 in the TB group, resulting in a post-intervention mean difference of 6.44 squats. The elevation growth rate of the TRX group was 23.75% higher than that of the TB group.

Final comparisons revealed a highly significant difference in test results between the two groups (p = 0.000 < 0.01, |d| = 1.417 > 0.8) (Fig. 9).

Figure 9 Comparison chart of balance ability between the two groups during the intervention process.

(where ** indicates a highly significant difference from the previous test result within the group, p < 0.01. ## indicates a highly significant difference within the group compared to the pre-intervention (E0), p < 0.01. && indicates a highly significant difference in the betweengroup comparison, p < 0.01).

Discussion

This study demonstrated that both 8-week TRX suspension training and traditional balance training improved surfers’ surf-specific balance abilities, but their efficacy diverged significantly after 5 weeks of intervention—a difference that persisted until the study conclusion. Specifically, TRX suspension training yielded superior improvements in surf-specific balance, confirming its targeted value for addressing the dynamic balance demands of surfing. This finding not only supports the utility of unstable training for sport-specific performance enhancement but also highlights TRX as a more effective modality than traditional stable-surface balance training for surfers.

The delayed emergence of performance differences (at week 5) provides insights into the time course of neuromuscular adaptation to unstable training. Neuromuscular adaptation—characterized by enhanced motor unit recruitment, improved proprioceptive feedback integration, and optimized intermuscular coordination (Wang et al., 2024)—is a gradual process that requires sustained exposure to specific stimuli. Traditional balance training, conducted on stable surfaces, primarily enhances static balance and basic postural control (Qiao & Yuan, 2010), which may explain the initial parallel progress between groups. In contrast, TRX training introduces continuous instability, forcing the central nervous system to recalibrate motor responses to constant perturbations—mirroring the unpredictable wave dynamics in surfing (Alcantara, Prado & Duarte, 2012). By week 5, TRX participants likely achieved sufficient adaptation to this stimulus, enabling more efficient activation of deep stabilizer muscles (e.g., transversus abdominis, multifidus) and refined proprioceptive mechanisms (Liu, Xu & Zhang, 2018), which are critical for maintaining balance during wave-riding maneuvers. This aligns with previous research showing that unstable training requires 4–6 weeks to induce measurable neuromuscular changes (Campa, Silva & Toselli, 2018), reinforcing the timeline observed in our study.

A key mechanism underlying TRX’s superiority lies in its ability to integrate strength, stability, and sensorimotor training under dynamic instability. Unlike traditional methods that isolate muscle groups (e.g., static balance board exercises), TRX employs closed-chain movements that engage the entire kinetic chain (Zheng & Qu, 2011). This holistic activation enhances core stiffness—a key determinant of postural stability during dynamic movements (Wang, Ding & Yin, 2016)—and promotes coordinated activation between proximal (core) and distal (limb) muscles, which is essential for executing surfing maneuvers like cutbacks and laybacks (Farley, Abbiss & Sheppard, 2017). Furthermore, TRX training’s oscillating balance challenges stimulate the vestibular and somatosensory systems more intensely than stable training, improving the speed and accuracy of postural corrections (Demirarar et al., 2021)—a capability directly transferable to adjusting to wave speed and direction changes. This mechanism may explain why TRX outperformed traditional training in our study, as surfing’s core demand is not just static balance but dynamic stability amid continuous perturbations.

Our findings align with broader literature on TRX’s efficacy in enhancing balance across sports (e.g., gymnastics: Li et al., 2010; taekwondo: Tian & Fu, 2024) but extend this knowledge by focusing on surf-specific balance. Notably, Tran et al. (2015) reported that unstable training did not significantly improve strength or sensorimotor skills in surfers, but this discrepancy likely stems from differences in training design and outcome measures: their study evaluated general strength, whereas ours focused on dynamic balance tailored to surfing’s movement patterns (e.g., lateral squatting, mimicking wave-side adjustments). This highlights the importance of aligning training modalities and assessment tools with sport-specific demands—a principle supported by Monaco et al. (2023), who emphasized that effective surfing training must replicate the biomechanics of wave riding.

The observed gender difference, with female surfers showing greater balance improvements from TRX training, warrants further discussion. While sample size limitations preclude definitive conclusions, this trend may relate to known gender differences in neuromuscular control: females typically exhibit greater reliance on proprioceptive feedback (vs. visual input) for balance, which TRX training specifically enhances. Additionally, hormonal factors (e.g., estrogen’s influence on muscle elasticity) may affect how females adapt to unstable stimuli. Future studies with larger cohorts should explore these mechanisms to inform gender-specific training protocols.

Practically, our results support integrating TRX into surfing training regimens at multiple levels. For elite surfers, TRX can break through plateaus in traditional balance training by targeting the neuromuscular adaptations most critical for wave riding—particularly during off-seasons when ocean access is limited. For beginners, TRX provides a controlled unstable environment to build foundational balance skills without the risk of wave-induced falls, accelerating their progression to on-water performance. Coaches can optimize TRX protocols by varying exercise complexity (e.g., progressing from static suspension to dynamic swinging) to match athletes’ skill levels, as suggested by Pang (2023) in aerobics training.

This study also validated the utility of the balance board lateral squat test as a surf-specific assessment tool. Unlike traditional balance tests (e.g., single-leg stance) that measure isolated static balance, this test integrates dynamic lateral movement, core engagement, and lower limb strength—mirroring the biomechanics of wave-side posture adjustments (Parsonage et al., 2017). Its sensitivity to TRX-induced improvements suggests it could serve as a valuable feedback tool for monitoring training efficacy, complementing on-water assessments.

Several limitations of this study should be acknowledged to guide future research. First, the current design focused exclusively on land-based training effects on balance ability, without directly verifying how these improvements translate to actual on-water surfing performance. While we demonstrated significant enhancements in static and dynamic balance—abilities widely recognized as foundational for surfing (Guo & Wang, 2022)—the critical step of linking land-based balance gains (e.g., balance board scores) to real-wave performance metrics (e.g., maneuver success rates, wave-riding duration) remains unexamined. Future studies should establish such correlations to confirm ecological validity.

Second, despite its utility, the balance board lateral squat test retains a static-dominant nature that may insufficiently capture the dynamic, multi-planar balance demands of wave surfing, where continuous adjustments to wave speed, amplitude, and direction are required (Farley, Abbiss & Sheppard, 2017). More ecologically valid assessments—such as tests conducted in simulated wave environments, water-based unstable platforms, or using wave simulators—would better reflect real surfing conditions and provide a more accurate evaluation of training efficacy.

Third, the 8-week intervention period, while sufficient to detect short-term improvements, limits conclusions about long-term effects. The durability of balance gains beyond this timeframe, as well as the potential for performance plateaus without protocol adjustments, remains unknown. Future research should include extended follow-up periods to assess the persistence of training effects, while also exploring optimal training frequencies, durations, and progressive intensity adjustments to sustain improvements over time.

Fourth, the relatively small sample size constrained our ability to account for variability across demographic and experience levels, including gender, age, and skill status (novice vs. expert). This limits the generalizability of our findings, particularly regarding subgroup-specific responses. Expanding sample sizes in future studies, combined with biomechanical assessment tools (e.g., force plates for center-of-pressure analysis, motion capture for kinematic tracking), would enable more objective and granular measurements of balance improvements across diverse populations.

Fifth, in terms of statistical rigor, this study relied primarily on traditional significance testing without incorporating multiple comparison corrections, which may increase the risk of Type I errors. Future work should adopt stricter statistical approaches, such as Bonferroni or Benjamini–Hochberg corrections, to enhance result reliability. Additionally, direct comparisons with other unstable training modalities (e.g., instability boards, BOSU balls) are lacking, leaving unanswered questions about TRX’s relative efficacy within the broader landscape of balance training methods.

Finally, it is important to clarify that this study focuses on exercise interventions for professional surfers, with a primary emphasis on athletic performance effects. Due to the unique complexity of surfing—shaped by variable wave conditions, competition rules, and environmental factors—no universally recognized standards (such as minimal clinically important differences, MCIDs) currently exist to quantify performance enhancements in this sport. Consequently, our analysis did not incorporate MCID frameworks. Instead, we employed scientifically rigorous experimental designs and measurement methods to evaluate intervention effects, with input from surfing professionals to ensure the ecological significance of results in real-world sports scenarios.

Conclusions

This study confirms that both 8-week TRX suspension training and traditional balance training effectively enhance the specialized balance abilities of surfers. However, significant differences in their efficacy emerged after 5 weeks of intervention and persisted until the end of the training period. TRX suspension training demonstrated superior effects in improving surf-specific balance, primarily due to its unique ability to replicate the dynamic instability of wave environments, promote integrated neuromuscular adaptations, and precisely target the core stability demands critical for wave riding. These attributes make TRX training more closely aligned with the specific physical requirements of surfing compared to traditional methods.

Given its proven efficacy, TRX suspension training can be recognized as a scientifically validated approach to enhance surfers’ specialized balance ability and is worthy of wider promotion and implementation in surfing-specific training programs. Practically, TRX equipment offers notable advantages such as cost-effectiveness, high durability, and long-term economic benefits, making it accessible for integration into daily training regimens. To optimize outcomes, coaches should design personalized TRX programs based on athletes’ skill levels and core strength, with flexible adjustments to training duration, intensity, and frequency.

For recreational surfers, whose primary focus is leisure, land-based TRX training can accelerate adaptation to balance control in unstable conditions, thereby enhancing their overall surfing experience. For elite surfers pursuing competitive excellence and high-precision technical execution, coaches are advised to progress training difficulty gradually—starting from simple static exercises to complex dynamic movements—and integrate TRX protocols with surfing-specific motor patterns. This integration can strengthen body control during actual wave riding, directly supporting technical refinement.

Regardless of skill level, surfers should prioritize gradual progression in TRX training and avoid prematurely attempting advanced maneuvers. When training independently, emphasis must be placed on movement quality, ensuring standardization and proficiency while developing the ability to perceive and regulate unstable environments. In summary, this study provides empirical support for incorporating TRX suspension training into surfing programs, offering coaches a science-backed tool to optimize athletes’ balance performance and technical precision in the unpredictable marine environment of competitive and recreational surfing.

Supplemental Information

Supplemental Information 1 Balance Board Lateral Squat

The authors warmly thank the athletes who participated in the intervention study; the athletes, coaches, teachers, and seniors who helped develop the intervention program and assisted with the experiment.

Additional Information and Declarations

Competing Interests

Author Contributions

Human Ethics

Data Availability

The authors declare there are no competing interests.

Zhaoyi Wang conceived and designed the experiments, performed the experiments, analyzed the data, prepared figures and/or tables, authored or reviewed drafts of the article, and approved the final draft.

Yong Ma conceived and designed the experiments, performed the experiments, analyzed the data, authored or reviewed drafts of the article, and approved the final draft.

Qian Huang conceived and designed the experiments, analyzed the data, authored or reviewed drafts of the article, and approved the final draft.

Zhihao Guo conceived and designed the experiments, authored or reviewed drafts of the article, and approved the final draft.

Mengyao Ja analyzed the data, prepared figures and/or tables, and approved the final draft.

Weitao Zheng conceived and designed the experiments, performed the experiments, analyzed the data, authored or reviewed drafts of the article, and approved the final draft.

The following information was supplied relating to ethical approvals (i.e., approving body and any reference numbers):

Wuhan Sports University granted ethical approval to carry out the study within its facilities (Ethical Application Ref: 2021030).

The following information was supplied regarding data availability:

The raw data is available in the Supplemental File.

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
