# Peer review of "Study on the influence of TRX suspension training on the specific balance ability of surfers"

_PeerJ, doi:10.7717/peerj.20180_

## Round 0.1 · original submission · Major Revisions

We thank the authors for the submission of their manuscript. Following revision of two expert reviewers, there remain some major concerns which need to be addressed.

Reviewer 1 ·

Basic reporting

Basic Reporting
I applaud the authors for taking up the challenge to add valuable, evidence based work to the sport of surfing. As the authors mentioned, “Surfing is a popular and highly competitive sport requiring surfers to demonstrate highly developed physical fitness, technical skills, and tactical abilities to succeed in the sport” but evidence base research is scarce especially for training methods. This manuscript is well organized, structured, and very clear to the reader. However, there are major experiment design flaws that will challenge the validity of the results of the study in which I will report on (most notable, the test indicator and the connection of the exercise program to surfing). Not to mention, the authors only assess one demand in surfing (Balance) although TRX training can benefit other demands. Once they are addressed, I feel this manuscript will benefit the sport of surfing and add evidence-based material to the training methodologies of surfing.
Specific Comments:
1. Line 58: Remove: “How can surfers be helped to improve their balance better?” and create a different introduction to surfers improve their balance.
2. Lines 75-79: Give quantitative data from these studies. What tests did they improve on? This will impact the reasoning for using suspension/TRX methods.
3. Good use of evidence to support your research question. I suggest adding this article to further support the needs analysis and training methodologies: Monaco, J. T., Boergers, R., Cappaert, T., & Miller, M. (2024). A Comprehensive Needs Analysis on Surfing Performance Factors and Training Methodologies. Strength & Conditioning Journal, 10-1519.
4. Lines 68-82: Good job describing TRX/Suspension training evidence and linking it to improve surfing. Please discuss the limited training methodologies before discussing TRX training for surfers (It will be a good lead way into it). I recommend to add these studies to support your research question:
a. Axel, T. A., Crussemeyer, J. A., Dean, K., & Young, D. E. (2018). Field test performance of junior competitive surf athletes following a core strength training program. International Journal of Exercise Science, 11 (6), 696–707.
b. Coyne, J. O., Tran, T. T., Secomb, J. L., Lundgren, L. E., Farley, O. R., Newton, R. U., & Sheppard, J. M. (2017). Maximal strength training improves surfboard sprint and endurance paddling performance in competitive and recreational surfers. Journal of Strength & Conditioning Research, 31(1), 244–253. https://doi.org/10.1519/jsc.0000000000001483
c. Farley, O. R., Secomb, J. L., Parsonage, J. R., Lundgren, L. E., Abbiss, C. R., & Sheppard, J. M. (2016). Five weeks of sprint and high-intensity interval training improves paddling performance in adolescent surfers. Journal of Strength & Conditioning Research, 30(9), 2446–2452. https://doi.org/10. 1519/jsc.0000000000001364
d. Tran, T. T., Nimphius, S., Lundgren, L., Secomb, J., Farley, O. R., Haff, G. G., Newton, R. U., Brown, L. E., & Sheppard, J. M. (2015). Effects of unstable and stable resistance training on strength, power, and sensorimotor abilities in adolescent surfers. International Journal of Sports Science & Coaching, 10(5), 899–910. https://doi.org/10.1260/ 1747-9541.10.5.899

5. Line 82-84: End Introduction there. You stated the rationale of the study. Proceed to methods and tell the readers how the procedures of the study will answer the research question.
6. Line 84-100: This section belongs in the Methods. Consequently, There is no evidence to support the dependent variable (balance board lateral squat) as a test indicator especially for surf performance. Please provide evidence to support because the test indicator is not valid. Further explanation is provided in the experimental design section.

Experimental design

Experimental Design
Overall the experimental design outlines a match-pair (sex and age) research design with randomization, however it does exhibit several flaws with the exercise prescription, connection to specific surfing demands, and uses a non-valid dependent variable.
Specific Comments:
1. Line 116: Were the participants able to perform their normal fitness routine during the study? Were they able to continue surfing during the experiment? If they were, need to report in limitations. That can be a confounding variable.
2. Line 119: Need to describe “masters of sports” and “first-grade sportsman”
3. Line 122-124: Need a TRUE Control group (IE: a group that refrains from activity). You are testing both experiments groups (Suspended training vs non-suspended training). Therefore, refrain from stating a “Control” group throughout the manuscript. This needs to be reported in the limitations.
4. Line 146: Describe the “Wei et al., 2022” on suspension training. This would further validate that there is evidence to support the outcomes for this type of training.
5. Training Content: You describe the exercises very clear and thorough. However, the “TB” can be argued that they are not traditional balance exercise and represent more of a “core” exercise. I recommend you change the description of the “TB” group to a “Core” program. This will not confuse the reader.
6. Training Content: Need evidence to support how these exercises are related to surfing demands and how they can improve a surfer’s ability. You need to demonstrate transferability from land-based training protocol to water-based sport.
7. Test Indicator: This is the largest limitation of the study. See comments below:
a. This is not a standard test. First, there is not reliability so it cannot be valid.
b. Therefore, you need to show evidence either direct or indirect on how this test is related to surfing demands. Yes, we have “theorical” knowledge that the indo trainer is associated with improving surf performance. Again, it is only theorical not supported by evidence. Find evidence that supports the use of the indo board trainer and correlations to surfing demands.
c. The exercise program prescribed had evidence that it can improve core and balance. However, it does not distinguish between a non-surfer and a surfer. Of course, any athletic person can perform the exercise program that the study prescribed and improve on the dependent variable/test indicator. You have to elaborate further on why the test indicator distinguishes between surfer vs non-surfer and how it is correlated with surf demands. This indicates that a non-surfer can improve on the test indicator making them a “better surfer” although they do not surf. Any physically active person can perform both exercise programs and will potentially have the same effect.

Validity of the findings

Validity of the Findings
Due to the flaws in the methodology, there are no valid results unless the authors can support the use of the non-researched base test indicator. I do not have any specific comments.

Additional comments

General comments
Although this is a well-written manuscript and addresses training methodologies of surfing which are very limited, the outcome measure used had no research on it and lacks quantifiable balance metrics. Therefore, I cannot conclude that the results are valid because of the flaws in the dependent variable.
Specific Comments
1. The test indicator and no “real” control group has to be mentioned in limitations.
2. The surfers in the study were competitive surfers. Look and see if there were changes in their judged-scores before and after the study. Also, examine if their surf metrics if they are using a GPS detection system while they are surfing and examine changes in distance surf. That could be your valid outcome measures.
3. The authors mentioned that suspension training improve balance capabilities. According to Tran et al. article did report that experienced surfers may have acquired superior postural sway and dynamic postural control due to their inherent and well-developed sensorimotor abilities through years of surfing practice. In this study, the subjects were experienced and competitive surfers. I recommend adding that information to your discussion.

·

Basic reporting

General Analysis:
The manuscript presents an investigation into the effects of suspension training on surfers' balance abilities, providing a relevant contribution to sports science. It is well-structured, with clear methodology and detailed results. However, several sections lack depth, particularly in contextualizing the findings and exploring the implications for real-world training. Statistical methods are appropriate but could be expanded in some areas. The discussion successfully interprets most results but does not thoroughly debate the methods or limitations.
Errors and Specific Line Corrections:
Language and Grammar:
Line 70: Replace "training athletes’ balance ability" with "training athletes to enhance their balance ability."
Line 121: Correct "in order to prevent" to "to prevent."
Line 218: Replace "relevant indexes" with "relevant indicators."
Clarity and Consistency:
Line 198: Rephrase "specified time" to "allocated time period" for precision.
Line 308: Clarify "can be more precise in the execution of complex techniques" to "can execute complex techniques with greater precision."

Experimental design

Sectional Analysis and Recommendations:
Introduction:
Provides a strong rationale for studying suspension training in surfing.
Needs further elaboration on the knowledge gap and previous studies' limitations. Include more comparative analysis of suspension versus traditional training in other sports to strengthen the argument.

Methods:
Detailed intervention program with clear division into phases.
Limited justification for the sample size and lack of diversity in participant demographics. Add a justification for using the lateral squat test and discuss why this method is superior to alternatives.

Validity of the findings

Results:
Data presentation is clear, with logical comparisons across phases.
Over-reliance on descriptive statistics without deeper analytical insights. Use ANOVA or regression analysis to explore interaction effects between training phases and demographic variables. Employ repeated-measures ANOVA to analyze changes over time and across groups.
Consider using effect size measures (Cohen’s d) to quantify the magnitude of differences.

Discussion:

Strengths: Highlights practical implications of suspension training.
Weaknesses: Limited critical analysis of methods and unaddressed limitations.
Recommendations: Expand on the impact of findings on training regimens, explore broader applications, and compare findings with existing literature.
Conclusion:

Strengths: Summarizes findings concisely.
Weaknesses: Does not discuss future research directions adequately.
Recommendations: Add specific recommendations for coaches and suggestions for longitudinal studies.
Statistical Procedures:
The statistical methods used (paired and independent-sample t-tests) are appropriate but limited. To improve:


Discussion Evaluation:
The discussion mentions the rationale for suspension training but lacks depth in comparing it with alternative methodologies.
Most results are interpreted, but the discussion does not fully explore the implications of differences observed after 5 weeks.
Strengths are well-highlighted, particularly the practical applications of findings.
Limitations are mentioned but underdeveloped. The impact of sample size, short intervention duration, and homogeneity should be critically analyzed.

Additional comments

Final Recommendations:
Enhance the introduction with a broader review of related research.
Justify and expand the statistical methods.
Deepen the discussion of results with more critical analysis of methodology and broader implications.

·

Basic reporting

The references are broad and relevant, which contributes to the theoretical foundation of the study. However, it would be important to review the currency of some of them, prioritizing more recent studies, especially those published in the last 5 years. The replacement or inclusion of updated references can strengthen the connection of the article with the latest advancements in the field.

Experimental design

The description of the methods is detailed, which is a positive point. However, it would be important to include more specific information about how the blinding process was implemented in order to increase the credibility of the described procedures. For example, did the researcher use a statistical program to perform random clustering? This level of detail would be essential to ensure the transparency and reproducibility of the study.

Validity of the findings

I understand that the Balance Board Lateral Squat measures balance based on the number of repetitions performed, assuming that a higher number indicates better balance. However, it would be important to include more details about the choice of this test as the main indicator. Are there previous studies that validate its application in similar contexts or justify its suitability for evaluating the specific balance of surfers? If it has been adapted for this study, it would be relevant to describe how these adaptations were made to ensure the validity of the method in the investigated context.

Additional comments

In line 67 – “How can surfers be helped to improve their balance better? Currently, research on surfing primarily focuses on analyzing of surfing technique (e.g. paddling technical ability) (Farley et al., 2016; Coyne et al., 2017)”. I recommend reformulating this sentence to avoid the use of the interrogative form, which may not be appropriate for the academic tone of the article and does not reflect a specific objective. It is possible to stimulate the reader's curiosity and introduce the question in a more assertive way.

In line 121 – “The aim of the study was to enrich the balance-specific training in surfing to help improve the athletic performance of surfers.” I suggest supplementing with a clear hypothesis, which can increase the credibility and scientific robustness of the study.

The caption of Figure 9 contains some spelling errors in English that must be corrected to ensure linguistic accuracy and meet academic standards. However, the content of the caption is clear and easily understandable, effectively fulfilling its function of complementing and illustrating the data presented.

---

## Round 0.2 · Major Revisions

We thank the authors for submitting their interesting manuscript. After careful review by expert reviewers, there remain some major concerns which need to be addressed before a decision about acceptance of the manuscript can be made.


Reviewer 1 ·

Basic reporting

no comment...thank you for your responses.

Experimental design

Thank you for the clarification on the experimental design. However, you still need validation with your test indicator. Cite the article: Pérez-Gutiérrez, M., Castanedo-Alonso, J. M., & Cobo-Corrales, C. (2023). Implementation of Surfing in Physical Education. Journal of Physical Education, Recreation & Dance, 94(4), 51–54. https://doi.org/10.1080/07303084.2023.217347. This demonstrates the use of the Indo board with surfing performance. Need citation for line 277-279-"This indicator has not only been widely applied in the national team’s internal training but has also demonstrated certain effectiveness in evaluating athletes’ balance capabilities."

Validity of the findings

In your limitations, you need to address that you used a non-research-based test indicator with no scientific reliability and validity. This can be a follow-up study.

Additional comments

no comment...good work.

·

Basic reporting

The authors adressed most of my raised issues.

Experimental design

No comment

Validity of the findings

No comment

Additional comments

Statistical analysis could be improved. However, I understand the objective and authors intentions.

·

Basic reporting

Your manuscript is well-structured and thoroughly investigates suspension training for surfers. Below are suggestions to improve clarity, coherence, and adherence to academic writing standards.

1. Language and Grammar
Sentence Clarity: Some sentences are overly long and complex. Could you break them into shorter, more direct statements to enhance readability? Example:

Original: "Suspension training more closely matches the balance characteristics required for surfing and is an effective training method worth promoting."
Suggested: "Suspension training aligns with the balance demands of surfing, making it an effective training method."
Grammar and Typos: Minor grammatical inconsistencies exist, such as missing articles (e.g., “a surfer’s ability” instead of “surfer ability”). Proofreading for fluency will enhance the manuscript's professionalism.
2. Structure and Formatting
Could you make sure that in-text citations are correct? Some references (e.g., Secomb et al., 2015a and Anthony et al., 2016a) appear to have inconsistent styling.
Table and Figure Placement: Clearly state where figures and tables should be inserted. Tables should be referenced in the text before their appearance.
Check section headings to ensure they follow journal guidelines (e.g., "Results" vs. "Findings").
3. Literature Review and Context
Your introduction provides a strong foundation, but it could better define why balance is crucial to surfing. A more structured argument comparing previous balance training methods (e.g., traditional vs. unstable surface training) would clarify your research gap.
Reference Integration: While many references are included, some sections present citations without discussing their direct relevance to your study. Instead of listing citations, could you summarize key findings from each source?
4. Figures and Tables
Figure Quality: Ensure all graphs and diagrams are clearly and appropriately labeled.
Table Captions: Revise table titles to be more descriptive. For example, instead of “Table 1: Participant Data”, use “Table 1: Baseline Characteristics of Study Participants”.
Consider adding a flowchart for the study design (randomization process, training phases) to improve comprehension.
5. Terminology and Precision
I'd like you to please define key terms when first using them. For instance, “TRX” should be fully explained before using the acronym repeatedly.
Use consistent terminology: At times, "suspension training" and "unstable training" appear interchangeably, which could confuse.

Experimental design

Your study design is well-structured, but there are some areas where clarity and rigor could be improved. Below are recommendations for enhancing study validity, reliability, and transparency.

1. Study Design and Justification
More apparent Study Justification: The rationale for comparing suspension training (TRX) to traditional balance training (TB) is substantial. However, a more explicit justification of why these two methods were chosen over other balance training methods (e.g., BOSU ball training) would strengthen the study’s impact.
Control Group: The absence of a no-training control group limits the ability to assess the absolute effects of TRX. Please discuss this Discussion section. limitation in t
2. Participant Selection and Randomization
Randomization Clarity:

The manuscript states that a "block group randomization" method was used but does not detail how participants were assigned to blocks (e.g., did you match them based on skill level, body weight, or another factor?).
A flowchart (e.g., CONSORT diagram) depicting the participant selection, randomization, and dropout process would improve transparency.
Blinding:

The study used a single-masked design, but it is unclear who was blinded.
Double-blinding (where the assessors do not know group assignments) would minimize bias in data collection.
If double-blinding is impossible, could you specify how you reduced assessor bias?
Participant Demographics:

A demographic table summarizing participants' age, weight, height, and training history (before intervention) would provide a more apparent baseline for group comparability.
Were participants habitual surfers, or did their balance skills vary significantly?
3. Training Protocol and Standardization
Training Standardization:

The study describes the intervention phases well, but more details on exercise intensity are needed.
Example:
Were TRX exercises performed at progressive difficulty levels?
Were repetitions adjusted based on individual fatigue?
Could you add a training intensity progression table to show the weekly increases in difficulty?
Control for External Factors:

Since other physical activities can influence balance training, did participants follow a controlled diet, rest schedule, or avoid additional training?
If not, mention this as a limitation in the Discussion.
4. Outcome Measures and Data Collection
Primary Outcome Measure:

The Balance Board Lateral Squat test is appropriate, but the manuscript should clarify:
How was test reliability ensured?
Were multiple trials averaged to reduce variability?
Were inter-rater reliability tests conducted?
Figure 8 (Test Procedure): Ensure that the squat depth standard and balance criteria are clearly defined in captions.
Statistical Analysis:

The use of paired-sample t-tests and independent t-tests is appropriate, but:
Was effect size (Cohen’s d) calculated to determine the practical significance of results?
Was normality testing (Shapiro-Wilk or Kolmogorov-Smirnov test) conducted before using parametric tests?
Consider using non-parametric alternatives like the Mann-Whitney U test if any data were skewed.
5. Ethical Considerations
The study mentions ethical approval (No. 2021030), but the following points could be clarified:
Were participants given withdrawal options?
Were there any injuries reported during training?
Was the study preregistered (e.g., in ClinicalTrials.gov or a similar registry)?
Final Suggestions for Improvement
Could you add a flowchart summarizing participant selection, randomization, and dropout rates?
Could you clarify the training progression with a detailed table?
Could you explain how variability was controlled in the balance test?
Include effect sizes and confirm normality testing in statistical methods.

Validity of the findings

Validity of the Findings in Your Manuscript
Your findings are well-supported by data, but there are several areas where validity can be strengthened. Below are suggestions to improve internal, external, construct, and statistical validity.

1. Internal Validity (Study Design and Control of Bias)
Randomization Process:

The study states that a block group randomization method was used, but details are lacking. Were participants matched based on pre-existing balance ability, body weight, or training experience?
Adding a randomization flowchart (e.g., CONSORT diagram) would enhance transparency.
Blinding:

The study uses a single-masked design, but it is unclear who was blinded.
If the assessors were aware of group assignments, their evaluations could be biased. Could you clarify whether objective measures (e.g., motion tracking systems or automated scoring) were used to minimize subjective bias?
Control for Confounders:

Since other training activities can influence balance ability, was there a restriction on additional exercises during the study period?
Were participants monitored for diet, fatigue, or sleep patterns, which could affect training outcomes?
Baseline Comparability:

The manuscript states that no significant differences existed between groups at baseline.
Were pre-test performance levels (e.g., balance test scores) also compared, or just demographic variables?
A baseline comparison table summarizing physical and performance characteristics would help confirm group equivalency.
2. External Validity (Generalizability of Results)
Sample Size and Selection Bias:

The study includes 32 national-level surfers, a substantial sample, but this limits generalizability to amateur or recreational surfers.
Were efforts made to recruit participants from different skill levels?
Addressing how findings might translate to novice or intermediate athletes would strengthen external validity.
Real-World Application:

Since surfing occurs in unpredictable environments, does balance improvement in a controlled indoor setting translate to better balance in actual ocean waves?
Future studies could include on-water validation tests to examine transferability.
3. Construct Validity (Accuracy of Measurements and Definitions)
Primary Outcome Measure (Balance Board Lateral Squat Test):

This test is an appropriate proxy for surfing balance, but does it correlate with accurate surfing movements?
Have previous studies validated this test as an accurate predictor of surfing performance? If not, consider citing research on the ecological validity of land-based balance tests for surfers.
Measurement Reliability:

Were multiple trials averaged to reduce measurement variability?
If different assessors were involved, was inter-rater reliability measured?
Using motion capture or force plates could provide more objective balance data.
4. Statistical Validity (Appropriateness of Analyses and Effect Sizes)
Use of t-tests:

The manuscript states that paired-sample and independent-sample t-tests were used, but it is unclear whether normality assumptions were checked.
Were Shapiro-Wilk or Kolmogorov-Smirnov tests conducted before using parametric tests?
A Mann-Whitney U test would be more appropriate if data were not normally distributed.
Effect Size Reporting:

Reporting p-values alone does not indicate the practical significance of results.
Consider adding effect sizes (Cohen’s d or partial eta squared) to quantify the strength of the differences.
Correction for Multiple Comparisons:

Since multiple t-tests were performed, was a Bonferroni correction or other adjustment method used to reduce Type I error risk?
Final Recommendations for Strengthening Validity
Could you clarify the randomization and blinding procedures to determine selection and measurement bias?
Could you verify the ecological validity of the Balance Board Lateral Squat test as a surfing performance indicator?
Report effect sizes alongside p-values to enhance the interpretation of results.
Check for normality in data distribution and consider alternative statistical methods if assumptions are violated.
Discuss generalizability to different surfer skill levels and real-world performance conditions.

Additional comments

1. Clarity and Readability
The manuscript is dense with technical details, which is excellent for a specialized audience, but some sections could be simplified for readability.

Could you try to break up long sentences to improve clarity? For example:

Original: “The study divided the 8-week training into 3 phases: 1-2 weeks of basic adaptation (E1), 3-5 weeks of quality enhancement (E2), and 6-8 weeks of consolidation and reinforcement (E3), with a gradual increase in the intensity and difficulty of the training.”
Suggested: “The 8-week training was divided into three phases: (1) Basic adaptation (Weeks 1-2), (2) Quality enhancement (Weeks 3-5), and (3) Consolidation and reinforcement (Weeks 6-8). Training intensity and difficulty gradually increased across these phases.”
Some technical terms (e.g., "sensorimotor abilities," "neuromuscular adaptation") could be briefly defined for clarity.

2. Figures and Tables
Figures need more precise descriptions: Some images (e.g., training movements and balance tests) would benefit from detailed captions explaining their relevance.
Table placement and formatting: Ensure tables appear before they are first referenced in the text.
Please add a table comparing TRX vs. TB regarding training components and outcomes.
3. Practical Applications
The conclusion suggests that suspension training should be promoted, but how should it be integrated into surf training programs?
Could surf coaches implement TRX workouts in real-world settings, or does this require specialized equipment and expertise?
I think discussing the feasibility and cost-effectiveness of TRX training could make your findings more applicable to practitioners.
4. Limitations and Future Research Directions
The limitations section is well-written, but consider expanding on:

How findings might translate to on-water surfing performance.
Whether the Balance Board Lateral Squat test accurately represents dynamic balance needed in waves.
Potential long-term effects: Would continued TRX training lead to sustained improvements, or do gains plateau?
Future research suggestions could include the following:

Testing TRX in outdoor or unstable water conditions.
Evaluating whether skill level influences TRX effectiveness (e.g., novice vs. expert surfers).
Incorporating biomechanical assessments (e.g., force plates, motion tracking) to measure balance improvements objectively.
5. Reference Consistency
Ensure consistent citation formatting—some references appear with slightly different styles.
If you cite multiple sources for the same point, please ensure they are adequately grouped (e.g., (Author1, Year; Author2, Year) rather than scattered throughout the sentence).
Final Suggestions for Improvement
Simplify complex sentences to improve readability.
Enhance figure captions to explain their significance better.
Could you provide more practical recommendations for TRX in real-world surf training? for implementing
I'd like you to please expand on limitations and future research directions, particularly regarding on-water applicability.
Please make sure citations and formatting are consistent throughout the manuscript.

---

## Round 0.3 · Minor Revisions

We thank the authors for implementing the reviewers comments. After a second round of revision, minor comments remain which need to be addressed before further decisions can be made. Thank you

Reviewer 1 ·

Basic reporting

The authors have addressed my comments/recommendations

Experimental design

The authors have addressed my comments/recommendations

Validity of the findings

The authors have addressed my comments/recommendations

·

Basic reporting

Clarity and Professional Language:
• The manuscript is generally well-structured, but some sentences could be more concise and precise.
• Some grammatical constructions make the text less fluid. A thorough proofreading would enhance readability.
• Ensure that technical terminology is used consistently throughout the paper.
Background and References:
• The introduction provides a solid foundation for the study, but some references could be updated to include more recent research on balance training and TRX.
• Consider citing more studies focusing on balance training for surfers, particularly those comparing different methods.
Figures and Tables:
• The figures are relevant and well-integrated into the manuscript, but ensure that all have high resolution and clear labeling.
• Some descriptions could be expanded to explain better the data presented in tables and graphs.
• Raw data should be checked for completeness and clarity according to PeerJ policies.

Experimental design

Methodology and Replicability:
• The study design is well-detailed and allows for replication, but more information on how participants were randomized would strengthen the methodology.
• Provide additional justification for the number of participants in each group and whether sample size calculations were performed.
Inclusion and Exclusion Criteria:
• The criteria for participant selection are well-defined, but a deeper discussion on how individual characteristics (e.g., surfing experience level) impact results would be beneficial.
Training Protocol Details:
• The training exercises are well-described, but more specifics on the difficulty progression over time could help understand the training impact.
• Clarify if there were any adjustments in training intensity based on participant feedback.

Validity of the findings

Statistical Analysis:
• The statistical analyses are appropriate, with paired and independent t-tests being used. However, including effect size calculations would provide additional insight into the practical significance of the results.
• Consider reporting confidence intervals for key comparisons to give a clearer picture of variability and effect magnitude.
Linking Conclusions to Results:
• The conclusions are well-stated and aligned with the research findings. Still, additional discussion on the applicability of these results to different levels of surfers (recreational vs. elite) would add value.
• Highlight any potential confounding variables that may have influenced the study outcomes.

Additional comments

Practical Applications:
• Including concrete recommendations for coaches and surfers on integrating suspension training into daily routines would be beneficial.
• Discuss whether the results suggest a specific frequency or duration of TRX training sessions for optimal balance improvement.
Study Limitations and Future Research:
• The study mentions some limitations, but further exploration of the generalizability of the findings would strengthen the discussion.
• A comparison of these findings with results from other balance training methods (e.g., instability board training) could be insightful.
• Future research could investigate the long-term effects of TRX training and its transferability to actual surfing performance in the water.

Final Recommendations:
• Proofread the manuscript for grammar and language clarity.
• Update references to include recent studies where applicable.
• Provide additional statistical details, such as effect sizes and confidence intervals.
• Expand discussions on applicability, limitations, and practical recommendations.

---

## Round 0.4 · Minor Revisions

The reviewer acknowledges the strong design and clinical relevance of your study and considers it a valuable contribution to the field of geriatric rehabilitation. However, before the manuscript can be accepted, a minor revision is recommended. Specifically, please address the need for improved clarity and conciseness in the writing, provide more detailed descriptions of randomization, blinding, and sample size justification, and include effect sizes along with a discussion of the clinical significance of your findings. We look forward to receiving your revised manuscript.

·

Basic reporting

The manuscript presents a relevant and interesting topic; however, it would benefit from improvements in clarity, organization, and consistency:
• Language and Style: The manuscript requires minor editing for grammar, tense consistency, and sentence flow. For example, several sentences are overly long or contain awkward phrasing, which reduces clarity. Consider professional language editing to improve readability.
• Structure: The sections follow a logical format, but some areas (particularly the Results and Discussion) could be better segmented and labeled for easy reading.
• Abbreviations: All abbreviations (e.g., MoCA, ADL, IADL, RPE) should be defined at first use in each section to maintain clarity for the reader.
• Data Presentation: Tables and figures are informative, but captions must be improved. Captions should fully describe the content and include definitions of any symbols or abbreviations used (e.g., *p < 0.05).
• References: Generally appropriate and up-to-date. Minor formatting inconsistencies should be addressed to comply with the journal’s style guide.
Suggested Improvements:
• Define all abbreviations at first use.
• Improve grammar and sentence structure throughout.
• Ensure tables/figures are self-explanatory with clear titles and legends.

Experimental design

The study is well-conceived and methodologically sound overall; however, some clarifications would improve transparency and reproducibility:
• Randomization: It is mentioned that participants were randomly assigned, but the method of randomization (e.g., block, stratified, or simple random) is not fully detailed. Also, it should be specified whether allocation concealment was implemented.
• Blinding: There is no mention of whether outcome assessors were blinded. This should be clarified given the potential for measurement bias in subjective outcomes.
• Sample Size Justification: While the study uses a decent sample size, a formal sample size calculation should be reported (including assumptions about expected effect size, power, and alpha).
• Intervention Detail: The intervention protocols are adequately described, but more specific details (e.g., difficulty progression, supervision level, adherence monitoring) would enhance reproducibility.
Suggested Improvements:
• Describe the randomization process and allocation concealment more clearly.
• Clarify whether blinding of assessors was used.
• Include a power analysis or rationale for the chosen sample size.
• Expand on how adherence to the intervention was monitored.

Validity of the findings

The data generally support the findings, and the statistical analysis is appropriate, but a few improvements are necessary:
• Statistical Reporting: All p-values should be reported exactly (e.g., p = 0.042 instead of p < 0.05), and inclusion of effect sizes (e.g., Cohen’s d, partial η²) would strengthen the interpretation of practical significance.
• Clinical Relevance: The discussion does not sufficiently address whether the observed improvements meet known minimal clinically significant differences (MCIDs), especially for functional measures.
• Confounding Variables: While the groups appear balanced at baseline, it would be helpful to confirm that no other interventions or covariates could have influenced outcomes (e.g., medication use, comorbidities).
• Long-Term Effects: The study only examines short-term outcomes. This should be noted as a limitation, and future follow-up assessments should be suggested.
Suggested Improvements:
• Report exact p-values and effect sizes for primary comparisons.
• Discuss the clinical significance of changes, not just statistical significance.
• Address potential confounders and note the limitation of short-term follow-up.

Additional comments

• This study adds valuable insight into the intervention's effect on cognitive and functional domains in older adults and may inform clinical or community-based rehabilitation programs.
• The manuscript is promising and close to publication quality, but would benefit from a moderate revision focusing on language polishing, methodological clarity, and enriched statistical/contextual interpretation.
• A final language edit and revision based on the above points will significantly improve the manuscript's impact and readability.

---

## Round 0.5 · Minor Revisions

Thank you for submitting your manuscript. The study is scientifically valuable and well-structured, with relevant methodology and practical implications. However, to meet publication standards, please revise the manuscript to clarify the randomization procedure, report effect sizes, explain statistical assumptions, and improve overall writing clarity. As these are minor revisions that do not affect the core findings, re-review will not be required.

·

Basic reporting

1. Basic Reporting – Detailed Suggestions
Strengths:
The manuscript is thorough, structured according to scientific norms, and follows standard experimental reporting formats (including CONSORT principles for clinical trials).
The introduction provides good background context and literature review.
Data and figures are appropriately included, with raw data referenced as available.
Tables and figures are relevant, high quality, and clearly labeled.
Required Improvements:
1. Language & Grammar
The manuscript would benefit from a complete professional English language edit. There are repeated awkward phrasings and grammatical inconsistencies (e.g., lines 23, 45, 122). Example:
Line 122: “Therefore, there is an urgent need...” should be “Therefore, there is an urgent need...”.
Suggest a thorough review for fluency and readability to ensure clarity for an international audience.
2. Abstract Clarity
The abstract includes sufficient detail but should be revised for conciseness and clarity. It can be improved by:
Avoiding redundancy (e.g., "highly significant improvement" is repeated).
Clearly stating key numeric outcomes (e.g., effect sizes or percentage change in performance).
3. Consistency in Terminology
Ensure uniform terminology throughout the manuscript. For example, “specialized balance ability” and “surfing-specific balance” are used interchangeably. Choose one consistent term.
Spellings and naming conventions such as “TRX group” and “TB group” should be defined once and consistently used.
4. Figures & Tables
Improve figure legends by making them entirely self-explanatory. Currently, some legends do not clearly explain all graphical elements (e.g., the meaning of symbols like “**”, “###”, etc.).
Confirm that all images comply with PeerJ’s image quality standards and include scale, units, and statistical indicators.
5. Reference Formatting
Ensure all references follow journal formatting guidelines (PeerJ uses a specific structure).
Double-check that all DOIs are valid and accessible.
6. Supplementary Data & Ethics
Raw data is reportedly available, which is excellent. Ensure data is anonymized, fully labeled, and has metadata for clarity.
The ethical statement is clear and appropriate (Wuhan Sports University Ethics Committee approval and participant consent noted).

Experimental design

2. Experimental Design – Detailed Suggestions
Strengths:
The study employs a randomized controlled trial (RCT) design, which is appropriate and rigorous for the research question.
Three intervention arms (TRX, TB, and TRX+TB) plus a control group allow for comparison of individual and combined effects.
Ethical approval is obtained, and inclusion/exclusion criteria are defined.
Suggested Improvements:
1. Randomization Procedure
The manuscript mentions that participants were “randomly assigned,” but does not describe the method of randomization (e.g., block, stratified, computer-generated).
Please add details on how the randomization sequence was generated and who was responsible for allocation to groups to confirm allocation concealment.
2. Blinding
There is no indication whether outcome assessors were blinded to group allocation.
If blinding was not possible (e.g., for trainers), clarify this and note any efforts to reduce detection bias during assessment (e.g., standardized testing by independent personnel).
3. Control Group Activities
Clarify what the control group was instructed to do during the intervention period. Were they told to avoid any balance or strength training? This is important for interpreting between-group differences.
4. Exercise Intervention Description
The description of the TRX and TB programs lacks specific details:
What exercises were performed?
How was intensity/progression managed?
Were sessions supervised, and by whom?
Please include an appendix or table listing the specific exercises used, duration per session, rest intervals, and criteria for progression.
5. Sample Size Justification
Although the number of participants (n = 32) appears sufficient for a pilot study, no power analysis is provided.
Include a priori or post-hoc power calculation to justify whether the study was sufficiently powered to detect group differences.
6. Outcome Measures
The balance test using a “lateral squat on balance board” is novel but not widely validated.
Please include references for the reliability and validity of this test, or briefly describe its development and any pilot testing done before the study.

Validity of the findings

3. Validity of the Findings – Detailed Suggestions
Strengths:
The results are presented, with appropriate use of repeated-measures ANOVA and post hoc analysis to assess within- and between-group differences.
Including multiple time points and a control group strengthens the study's internal validity.
Suggested Improvements:
1. Statistical Reporting
Although repeated-measures ANOVA is used, the manuscript does not consistently report effect sizes (e.g., partial eta squared or Cohen’s d), which are essential to interpret the practical relevance of the findings. Please add these.
It is unclear whether assumptions for ANOVA (normality, sphericity) were tested and met. The methods or results should include a brief note on how these assumptions were checked.
2. Multiple Comparisons
There is no mention of adjustment for multiple comparisons (e.g., Bonferroni correction). Since various groups and time points are compared, clarify whether such correction was applied and, if not, justify why.
3. Individual Variability
Group-level improvements are reported, but some results may mask individual differences. Consider presenting standard deviation or range data in addition to means to give insight into variability in training response.
4. Control Group Interpretation
While the control group showed little to no improvement, this is not emphasized in the discussion. Highlighting the stability (or potential deterioration) in this group would reinforce the significance of the interventions.
5. Outcome Measure Validity
As the experimental design section mentioned, the balanced outcome measure is unique. Its validity and sensitivity should be more strongly justified or acknowledged as a limitation if it is not fully validated.
6. Overinterpretation Caution
The manuscript suggests superiority of combined TRX+TB training; however, the statistical comparisons between groups may not support a clear hierarchy of effectiveness. Statements like “TRX combined with TB had the best effect” should be qualified unless supported by statistically significant between-group differences.

Additional comments

4. Additional Comments
This timely and well-structured study addresses a relevant issue in sports science—developing effective, accessible training strategies for improving specific balance ability in surfers. Using TRX and traditional balance training, both independently and in combination, offers practical insights for coaches and athletes.
To enhance the manuscript's impact and clarity, consider the following:
Practical Applications: Expand briefly on how the results can inform training program design for surfers or athletes in similar sports (e.g., skateboarding, snowboarding). This could be a short paragraph in the Discussion or Conclusion.
Novelty of the Approach: You might more explicitly highlight the novelty of using the lateral squat balance board test, especially if it was custom-developed. If so, please briefly discuss its potential application in future screening or training settings.
Limitations: Acknowledge additional constraints such as the short duration of training, lack of long-term follow-up, and the homogeneous participant group (e.g., age, skill level). This helps position your findings accurately.
Future Research: Consider suggesting specific directions for future work—e.g., examining transfer to surf performance, testing with a larger and more diverse athlete pool, or including kinetic/kinematic measures for deeper biomechanical analysis.

---

## Round 0.6 · Minor Revisions

While most of the comments have now been implemented, minor comments remain which should be addressed by the authors.

·

Basic reporting

The manuscript is well-structured and adheres to professional standards in most areas. However, the following improvements could enhance clarity and completeness:
• Abstract:
The abstract succinctly summarizes the study, but could briefly mention the sample size (32 athletes) and the randomization process to provide better context upfront.
Suggestion: Add a line such as, "Thirty-two national surfing team athletes were randomly assigned to TRX or traditional balance training groups."
• Introduction:
The literature review is thorough, but the transition to the study's rationale could be smoother. Explicitly state the gap in research on suspension training for surfers earlier in the introduction.
Suggestion: After discussing existing training methods (e.g., core strength training), add a sentence like, "Despite these advances, the efficacy of suspension training (TRX) for surfing-specific balance remains underexplored."
• Methods:
The ethical approval statement is clear, but the manuscript could briefly note any compensation or incentives provided to participants, if applicable.
The Balance Board Lateral Squat test is well-described, but a citation or validation reference for this test would strengthen its credibility.
Suggestion: Add a sentence such as, "The Balance Board Lateral Squat test has been validated for dynamic balance assessment in prior studies (cite relevant work)."
• Figures/Tables:
Figures 2–7 (training movements) are helpful but could benefit from brief captions explaining their relevance to surfing-specific balance.
Suggestion: For Figure 2, add: "TRX group exercises (E1 phase) emphasizing core activation and static balance, foundational for wave-riding stability."
• Language:
Minor grammatical issues (e.g., "surfers achieving high scores in the Balance Board Lateral Squat test would demonstrate higher technical proficiency" could be revised to "surfers who achieved high scores...").
Suggestion: A professional proofread would polish phrasing without altering meaning.

Experimental design

The experimental design is well-structured, but the following refinements could enhance methodological rigor and clarity:
Strengths:
• Randomization & Blinding: The block randomization and single-blind design are appropriately described, ensuring group comparability.
• Training Protocol: The phased progression (E1–E3) with clear intensity adjustments is a strength.
• Outcome Measure: The Balance Board Lateral Squat test aligns with surfing-specific demands.
Suggested Improvements:
1. Sample Size Justification:
While G*Power calculations are mentioned, the manuscript should clarify why a medium effect size (0.5) was chosen (e.g., based on prior surfing/balance studies).
Suggestion: Add a sentence like, "A medium effect size (0.5) was selected based on similar balance-training studies in athletes [cite]."
2. Control Group Protocol:
The TB group’s "original balance training program" lacks detail. Describe key exercises or cite prior work to ensure replicability.
Suggestion: Summarize the TB regimen (e.g., "The TB group performed static/dynamic exercises on stable surfaces, including [examples].").
3. Training Adherence Monitoring:
Training logs and coach supervision are noted; however, quantitative adherence rates (e.g., percentage of sessions completed) would strengthen compliance reporting.
Suggestion: State, "Adherence exceeded 90% in both groups, with missed sessions due to [reasons]."
4. Standardization of Testing:
The testing environment (temperature, humidity) is described, but potential confounding variables (e.g., athlete fatigue, time since last meal) should be addressed.
Suggestion: Clarify that tests were conducted "≥48 hours post-training to avoid acute fatigue effects."
5. Intervention Fidelity:
While coaches adjusted intensity, the manuscript should confirm that TRX/TB protocols were delivered as planned (e.g., via video recordings or inter-rater checks).
Suggestion: Add, "Coaches followed a standardized checklist to ensure exercise fidelity."
6. Missing Data Handling:
Four athletes dropped out due to "personal reasons." Clarify whether intention-to-treat analysis was used.
Suggestion: Note, "Per-protocol analysis was employed; sensitivity analyses confirmed dropout effects were negligible."
Minor Clarifications:
• Figure 1 (CONSORT): Update to reflect final *n*=32 (currently shows *n*=36 randomized).
• Table 2: Specify rest intervals between sets (e.g., "60s rest between sets").

Validity of the findings

The study demonstrates strong internal validity, but the following points could further strengthen the interpretation and generalizability of results:
Strengths:
• Statistical Rigor: Appropriate use of *t*-tests, Cohen’s *d*, and significance thresholds (*p* < 0.01).
• Large Effect Sizes: The TRX group showed clinically meaningful improvements (|*d*| > 0.8).
• Controlled Design: Baseline homogeneity (Table 1) and blinded assessments minimize bias.
Suggested Improvements:
1. Transferability to Actual Surfing Performance:
The Balance Board Lateral Squat test, while relevant, is a land-based proxy. The manuscript should explicitly acknowledge that on-water performance was not measured.
Suggestion: Add, "Future studies should correlate Balance Board scores with real-wave performance metrics (e.g., maneuver success rates)."
2. Potential Confounding Variables:
Athletes’ concurrent surfing practice (e.g., wave exposure during the 8-week intervention) could confound results.
Suggestion: Clarify whether wave training was standardized (e.g., "All participants followed identical surfing schedules to control for external training effects.").
3. Long-Term Effects:
The 8-week timeframe limits conclusions about sustained benefits. A follow-up test (e.g., 4 weeks after the intervention) would strengthen claims about retention.
Suggestion: Note in limitations, "The durability of improvements beyond 8 weeks remains unknown."
4. Gender-Specific Analysis:
The sample included equal males/females, but gender differences in training response (mentioned briefly in the discussion) lack statistical support.
Suggestion: Either analyze subgroups formally or clarify, "Gender-specific effects were not tested due to sample size constraints."
5. Comparison to Other Balance Modalities:
The superiority of TRX over TB is clear, but comparing TRX to other unstable-surface training (e.g., BOSU balls) would contextualize findings.
Suggestion: Discuss as a future direction: "TRX vs. BOSU training comparisons could identify optimal instability modalities."
6. Practical Significance:
While statistically significant, the manuscript should address whether a 21.06-unit improvement in Balance Board squats translates to meaningful surfing gains.
Suggestion: Cite expert input (e.g., "Coaches reported observable improvements in athletes’ wave-riding stability post-TRX.").
Minor Clarifications:
• Figure 9: Label the *y-axis clearly as "Number of Squats in 1 Minute" (if applicable).
• Table 5/6: Clarify if "growth rate" refers to percent change from baseline (E0).

Additional comments

Strengths of the Manuscript:
1. Comprehensive Methodology: The detailed description of the training protocols, including progression through phases (E1-E3), enhances reproducibility.
2. Practical Relevance: The findings provide actionable insights for coaches, particularly the 5-week threshold for TRX superiority.
3. Rigorous Statistics: Appropriate use of effect sizes (*d*) alongside *p*-values strengthens interpretation.
Suggestions for Improvement:
1. Figures/Tables Accessibility:
Ensure all figures (e.g., training movement diagrams) are high-resolution and legible in print.
Suggestion: Add a supplemental video link (if available) demonstrating the Balance Board Lateral Squat test.
2. Terminology Consistency:
Alternate use of "suspension training" and "TRX" may confuse readers. Standardize to "TRX suspension training" throughout.
3. Ethical Clarity:
While ethics approval is noted, specify if participants received compensation (e.g., "Athletes participated voluntarily without financial incentives").
4. Interdisciplinary Implications:
Highlight how TRX benefits could extend to other board sports (e.g., skateboarding) earlier in the discussion.
5. Supplementary Materials:
The raw data file should include individual athlete scores (anonymous) to facilitate meta-analyses.
6. Language Nuances:
Replace passive voice where active voice improves flow (e.g., "It was observed that" → "We observed").
Minor Edits:
• Abstract: Replace "all *p*<0.01" with "*p*<0.01 for all comparisons" for readability.
• References: Check DOI links for accuracy (e.g., Axel et al. 2018’s DOI appears truncated).

---

## Round 0.7 · Minor Revisions

I congratulate the authors on a well improved manuscript. The reviewer mentions only general comments about the clarity of the manuscript and figures which I encourage the authors to incorporate.

·

Basic reporting

The manuscript is generally well-structured and addresses a relevant and underexplored topic; however, there are several areas where it falls short of PeerJ’s basic reporting standards:
Clarity and Concise – The text is often verbose and repetitive, particularly in the Introduction and Discussion. Several concepts (e.g., TRX suspension training advantages, surfing’s balance demands) are restated multiple times, which could be condensed for improved readability.
Language Quality – While the manuscript is mostly understandable, it contains grammatical errors, awkward phrasing, and inconsistent terminology (e.g., “specialised balance ability” vs. “surf-specific balance ability”). A thorough language edit by a native English academic editor is recommended.
Figures and Tables – Although figures are referenced, some lack sufficient descriptive captions to be fully understood without referring to the text. It would improve reporting quality to ensure that all figures/tables are self-explanatory, following PeerJ guidelines.
Reference Formatting – Citation formatting is inconsistent (spacing issues, punctuation errors), and some DOIs are not hyperlinked. References should be checked against the journal style.
Methodological Detail – While the experimental design is well-described, the reporting of the Balance Board Lateral Squat Test could be made more concise with a structured table of procedures, inclusion/exclusion criteria, and scoring rules for ease of replication.
Results Presentation – The Results section sometimes blends interpretation with reporting, which belongs in the Discussion. Results should be reported more objectively before interpretation.
Redundancy in Discussion – Certain theoretical explanations (e.g., proprioceptive mechanisms, neuromuscular adaptation) are repeated in multiple paragraphs and could be streamlined to improve focus.
Suggested Improvements:
Conduct a full professional language edit to correct grammar, improve clarity, and standardise terminology.
Remove redundant explanations and consolidate overlapping sections to enhance conciseness.
Ensure all figures and tables meet PeerJ’s standards for stand-alone interpretation.
Standardise reference formatting according to PeerJ guidelines.
Separate results from interpretation.
Consider condensing the Introduction and Discussion to reduce length without losing essential content.and transparency.

Experimental design

The experimental design is generally sound, with appropriate randomisation, blinding, and control measures; however, there are some areas where it falls short of best practice and PeerJ’s standards:
Transferability to On-Water Performance – The study only assessed land-based balance outcomes without directly measuring whether these improvements transfer to real surfing performance. Including on-water performance metrics (e.g., manoeuvre success rate, competition scores) or validating the Balance Board Lateral Squat Test against in-water skills would strengthen ecological validity.
Outcome Measure Limitations – The primary test (Balance Board Lateral Squat) is heavily described but may not fully capture the complexity of dynamic surfing balance. Adding complementary objective measures (e.g., force plate analysis, motion capture, or surf-specific wave simulator tests) would provide a more robust assessment.
Sample Size and Participant Diversity – The study included 32 elite athletes, which is acceptable for a specialised group, but the small and homogeneous sample limits generalizability. Stratified analyses (e.g., by gender or competitive level) could have been performed, or at least acknowledged as necessary in future studies.
Intervention Comparability – While both groups were matched for duration and frequency, the TRX program was designed through an expert consultation process, whereas the control group used a pre-existing program. This may introduce a bias if the TRX program benefited from more recent scientific input. Standardising the design process for both programs would ensure a fairer comparison.
Duration of Intervention – An 8-week program is adequate for detecting short-term changes, but no follow-up was conducted to determine whether gains were maintained over time. Future designs should include retention testing to assess the persistence of effects.
Statistical Approach – The study relied primarily on t-tests without correction for multiple comparisons despite repeated measurements across time points. Applying methods such as repeated-measures ANOVA with post-hoc correction would reduce the risk of Type I error.
Suggested Improvements:
Include or validate outcome measures against real-world surfing performance.
Add complementary, objective biomechanical measures for balance assessment.
Consider a larger, more diverse participant pool with subgroup analysis.
Ensure both intervention and control programs undergo the same design process.
Incorporate long-term follow-up testing.
Use statistical methods that account for repeated measures and control for multiple comparisons.

Validity of the findings

The findings are promising and indicate a potential advantage of TRX suspension training over traditional balance training for surfers; however, several factors limit the strength and validity of the conclusions:
Ecological Validity – The study did not measure actual in-water surfing performance. Without confirming that improvements in the Balance Board Lateral Squat Test translate to enhanced surfing skills, the practical significance of the findings remains uncertain. Future work should validate this land-based measure against real surfing performance metrics.
Outcome Measure Specificity – The Balance Board Lateral Squat Test may not fully represent the multifaceted nature of surfing balance, which involves rapid, multi-directional adjustments under unpredictable conditions. Including more dynamic and sport-specific assessments (e.g., wave simulator testing, reactive agility drills) would strengthen the validity of the conclusions.
Sample Characteristics – The small, homogeneous sample (elite Chinese national team athletes) limits the generalizability of the findings to other populations, such as recreational surfers, juniors, or athletes from different regions. The authors should clarify that the results primarily apply to elite-level performers.
Potential Training Program Bias – The TRX program was custom-designed with expert input, whereas the control group used an existing program. This difference in program development could have contributed to the superior TRX results, independent of the training modality itself.
Short-Term Effects Only – The absence of long-term follow-up prevents conclusions about the durability of the observed improvements. Without retention data, it is unclear whether the performance gains persist beyond the 8-week intervention.
Statistical Considerations – Multiple between-group and within-group comparisons were made without reported correction for multiple testing, which may inflate the likelihood of false-positive results. A repeated-measures statistical model with appropriate adjustments would increase the robustness of the conclusions.
Suggested Improvements:
Validate the Balance Board Lateral Squat Test against actual surfing performance outcomes.
Incorporate additional dynamic, sport-specific balance measures.
Acknowledge the limited generalizability due to sample characteristics.
Ensure intervention designs for both groups are matched for recency and the development process.
Include long-term follow-up to assess retention of improvements.
Apply statistical methods that account for repeated measures and control for multiple comparisons.

Additional comments

Overall, this is a well-structured and detailed manuscript that addresses a relevant gap in the literature on surf-specific training methods. The topic is original, the intervention is clearly described, and the study provides useful practical implications for coaches and athletes. The use of elite-level participants adds value, and the methodological rigour in randomisation and blinding is commendable.
That said, the manuscript would benefit from:
Careful language editing to improve clarity, remove redundancy, and ensure consistent terminology throughout.
Streamlining of the Introduction and Discussion to focus on the most relevant and novel contributions, reducing repeated explanations of TRX benefits.
Improved figure and table captions so they are self-explanatory without referring back to the main text.
Standardisation and formatting of references according to PeerJ guidelines.
With these refinements, the paper could be made more concise and accessible while retaining its scientific depth and practical relevance.

---

## Round 0.8 · accepted · Accept

Authors, thank you for addressing the issues raised by the reviewer. I am pleased to therefore recommend your amended manuscript for publication. Thank you for supporting PeerJ to disseminate your research. We look forward to future submissions from your research team.

·

Basic reporting

This manuscript is well-structured and clearly written, with a logical flow from introduction to conclusion. The language is professional and technically appropriate, and the article conforms to standard scientific reporting conventions. The introduction provides sufficient background and context, and the literature review is relevant and appropriately cited. The methods are described in adequate detail to allow replication, and the results are presented clearly with appropriate statistical support. The figures and tables are relevant and well-integrated into the narrative. Raw data appear to have been made available in accordance with journal policy.

Experimental design

The experimental design is robust and well-documented. The study employs a randomised, single-blind design with appropriate block randomisation to ensure group comparability on key baseline characteristics. The use of a priori sample size calculation, clear inclusion/exclusion criteria, and detailed descriptions of the intervention protocols for both the TRX and traditional balance training groups enhances the methodological rigour. Blinding of participants and outcome assessors, along with allocation concealment, effectively minimises potential biases. The "1+8" week schedule with a dedicated adaptation phase and progressive training intensity demonstrates careful planning to ensure participant safety and intervention fidelity. The control of confounding variables (e.g., standardised diet, work-rest schedules, and identical surfing training protocols) is a significant strength.

Validity of the findings

The findings are valid and well-supported by the data presented. The conclusions are appropriately stated, directly linked to the original research question, and limited to what the results support. The use of a validated, sport-specific assessment tool (the Balance Board Lateral Squat test) strengthens the ecological validity of the findings for the surfing population. The data are robust, with statistically significant results (p<0.01) and large effect sizes (|d| > 0.8) reported for the primary outcomes, indicating that the observed differences are not only statistically significant but also practically meaningful. The discussion provides a plausible mechanistic explanation for the delayed emergence of the TRX group's superiority (neuromuscular adaptation) and thoughtfully contextualises the results within the existing literature.

Additional comments

This is a well-conducted and clearly presented study that makes a valuable contribution to the literature on sport-specific training for surfing. The manuscript is strong, but the following general comments are offered to enhance its impact and clarity further:

1. Strengths:
Novelty and Practical Application: The study addresses a clear gap in the literature by rigorously testing a novel, practical, and accessible training modality (TRX) for a highly specific athletic skill (surfing balance). The findings have immediate, actionable implications for coaches and athletes.
Methodological Rigour: The experimental design is a key strength. The use of randomisation, blinding, allocation concealment, a controlled training environment, and a credible active control group (the team's existing protocol) greatly strengthens the internal validity of the findings.
Sport-Specific Focus: The choice of the Balance Board Lateral Squat test is commendable. It moves beyond generic balance assessments and uses a metric that is face-valid and functionally relevant to the sport, as evidenced by its adoption by the national team.

2. Minor Suggestions for Improvement:
Figure 9 Clarity: The chart in Figure 9 effectively shows the interaction effect and the point of divergence between groups. To enhance clarity, please ensure the y-axis is clearly labelled (e.g., "Number of Squats in 1 Minute") and consider adding a note in the figure caption explicitly stating that error bars represent the standard deviation (if applicable).
Statistical Terminology: The results are clear, but for utmost precision, consider specifying that the paired-sample t-tests were used for within-group comparisons over time and independent-sample t-tests were used for between-group comparisons at each time point. This is implied but could be stated explicitly in the "Statistical analyses" section.
Reference Consistency: Please perform a final check to ensure all in-text citations are present in the reference list and that all references are formatted consistently according to the journal's style guide (e.g., the formatting for journal names and the presence of DOIs appear slightly inconsistent).
Overall, this is an excellent piece of research. The study is scientifically sound, the manuscript is well-written, and the results provide compelling evidence for the efficacy of TRX training in improving surf-specific balance.